# Optimization of a Luteolin-Loaded TPGS/Poloxamer 407 Nanomicelle: The Effects of Copolymers, Hydration Temperature and Duration, and Freezing Temperature on Encapsulation Efficiency, Particle Size, and Solubility

**DOI:** 10.3390/cancers15143741

**Published:** 2023-07-24

**Authors:** Muhammad Redza Fahmi Mod Razif, Siok Yee Chan, Riyanto Teguh Widodo, Yik-Ling Chew, Masriana Hassan, Shairyzah Ahmad Hisham, Shamima Abdul Rahman, Long Chiau Ming, Ching Siang Tan, Siew-Keah Lee, Kai Bin Liew

**Affiliations:** 1Faculty of Pharmacy, University of Cyberjaya, Cyberjaya 63000, Malaysia; redza.razif@gmail.com (M.R.F.M.R.); shairyzah@cyberjaya.edu.my (S.A.H.); shamima@cyberjaya.edu.my (S.A.R.); 2School of Pharmaceutical Science, Universiti Sains Malaysia, Gelugor 11800, Malaysia; sychan@usm.my; 3Faculty of Pharmacy, University Malaya, Kuala Lumpur 50603, Malaysia; riyanto@ummc.edu.my; 4Faculty of Pharmaceutical Science, UCSI University, Kuala Lumpur 56000, Malaysia; chewyl@ucsiuniversity.edu.my; 5Faculty of Medicine and Health Sciences, Universiti Putra Malaysia, Serdang 43400, Malaysia; masriana@upm.edu.my; 6School of Medical and Life Sciences, Sunway University, Bandar Sunway 47500, Malaysia; longchiauming@gmail.com; 7School of Pharmacy, KPJ Healthcare University College, Nilai 71800, Malaysia; 8M. Kandiah Faculty of Medicine and Health Sciences, Universiti Tunku Abdul Rahman, Kajang 43000, Malaysia; leesiewkeah@utar.edu.my

**Keywords:** luteolin, micelle, film hydration method, freeze-drying temperature, hydration temperature

## Abstract

**Simple Summary:**

Luteolin is a natural compound from plants with various medicinal benefits, including anti-cancer activities. However, luteolin has limited use clinically due to its low solubility and absorption into the body. Hence, the objective of this research work was to formulate and optimize a luteolin nanomicelle formulation with improved solubility. The nanomicelle was prepared using D-α-tocopheryl polyethylene glycol 100 succinate (TPGS) and poloxamer (Pol). The solubility of Lut-loaded micelles increased up to 459-fold compared to pure Lut in water. The release study showed that Lut-loaded micelles exhibited sustained release behavior. The effects of copolymers, hydration temperature and duration, and freezing temperature were studied to find out the optimum formulation of a luteolin–micelle complex. This study has demonstrated that several factors need to be considered when developing such nanoparticles in order to obtain a well-optimized micelle. The formulation has potential to be utilized in cancer therapy.

**Abstract:**

Background: Luteolin is a flavonoid compound that has been widely studied for its various anti-cancer properties and sensitization to multidrug-resistant cells. However, the limited solubility and bioavailability of Lut hindered its potential clinical use. Theoretically, the combination of this compound with vitamin E TPGS and poloxamer 407 can produce a synergistic effect to enhance tumor apoptosis and P-glycoprotein inhibition. This study aimed to develop and optimize vitamin E TPGS/Poloxamer 407 micelles loaded with luteolin through investigating certain factors that can affect the encapsulation efficiency and particle size of the micelle. Methods: A micelle was prepared using the film hydration method, and the micellar solution was lyophilized. The cake formed was analyzed. The factors investigated include the concentrations of the surfactants, ratio of vitamin E TPGS/Poloxamer 407, temperature of the hydrating solution, duration of hydration, and freezing temperature before lyophilization. The effects of these factors on the encapsulation efficiency and particle size of the micelle were also studied. The encapsulation efficiency was measured using a UV-Vis spectrophotometer, while particle size was measured using dynamic light scattering. Results: The optimized micelle was found to have 90% encapsulation efficiency with a particle size of less than 40 nm, which was achieved using a 10% concentration of surfactants at a vitamin E TPGS/Poloxamer 407 ratio of 3:1. The optimized temperature for hydrating the micellar film was 40 °C, the optimized mixing time was 1 h, and the optimized freezing temperature was −80 °C. The solubility of the luteolin-loaded micelles increased 459-fold compared to pure Lut in water. The critical micelle concentration of the vitamin E TPGS/Poloxamer 407 micelle was 0.001 mg/mL, and the release study showed that luteolin-loaded micelles exhibited sustained release behavior. The release of luteolin from a micelle was found to be higher in pH 6.8 compared to pH 7.4, which signified that luteolin could be accumulated more in a tumor microenvironment compared to blood. Conclusion: This study demonstrated that several factors need to be considered when developing such nanoparticles in order to obtain a well-optimized micelle.

## 1. Introduction

Breast cancer (BC) is the most diagnosed cancer, causing about 0.7 million deaths worldwide in 2020 [1,2]. BC is mostly treated with surgery, chemotherapy, hormone therapy, biological therapy, radiation therapy, and phototherapy [3,4,5]. Adverse effects that are commonly observed with current chemotherapies include nausea, vomiting, weight gain, hair loss, and an increased chance of infection [6]. These occur since most of the conventional chemotherapeutic drugs that inhibit rapidly growing cancer cells also attack other rapidly growing normal cells such as gastrointestinal cells, bone marrow cells, and hair cells [7].

Multidrug resistance (MDR) is one of the major challenges in treating cancer. Chemotherapy is the most common treatment for BC, and statistical data has shown that over 90% of the mortalities of cancer patients are attributed to failed conventional chemotherapeutic drugs due to drug resistance [8]. ATP-binding cassette (ABC) proteins such as P-glycoprotein (P-gp) are responsible for the protection of cancer cells from high concentrations of cytotoxic drugs causing elevated effluxes of the drugs from the cancer cells. P-gp is highly expressed on the surfaces of the endothelial cells of cancer cells, which contributes to the lower penetration of cytotoxic drugs [9,10]. Due to that, conventional cytotoxic drugs have very low efficacy and are least favorable when cancer cells become resistant.

The search for effective treatments and preventive measures led researchers to explore the role of natural bioactive compounds in cancer management. Bioactive compounds are molecules found in various natural sources, including plants, animals, and microorganisms, which have physiological benefits and can potentially reduce the risks of certain diseases, including cancer [11]. One of the few groups of bioactive compounds that have received significant attention in cancer research is flavonoids. In in vitro and in vivo studies, flavonoids exhibited anti-proliferation, anti-metastatic, and immuno-modulatory properties. Despite flavonoids’ well-known antioxidant functions, recent studies found that they also interact directly with proteins, making them an ideal small molecule for modulating enzymes, transcription factors, and receptors. These unique flavonoid qualities suggest new ways to alter tumor signaling, address chemo-resistance, and retrain the tumor microenvironment [12,13].

Luteolin (Lut) is a flavonoid. Lut is frequently present in fruits, vegetables, and medicinal herbs such as green peppers, celery, broccoli, and parsley [14]. Recent research has shown that Lut has a variety of biological effects, primarily because of its antioxidant and free-radical-scavenging properties, including anti-inflammation, anti-allergy, and anticancer effects [15,16]. Furthermore, Lut has been found to reverse MDR and desensitize chemo-resistant cells when co-administered with other chemotherapeutic drugs such as oxaliplatin [17]. Samy et al. [18] reported that Lut caused cytotoxicity in cancer cells but not in normal cells. However, Lut has some limitations such as poor solubility, poor bioavailability, and low oral absorption, preventing Lut from reaching its full potential in clinical applications. Therefore, it is important for researchers in the field to find solutions to its limitations.

Nanoparticles have emerged as promising delivery systems for hydrophobic drugs, addressing the challenges of their poor solubility and bioavailability. These nanoparticles can encapsulate hydrophobic drugs, protecting them from degradation and facilitating their delivery to target sites [19]. Various types of nanoparticles, such as polymeric micelles, solid lipid nanoparticles, noisome, and phytosome, have been proven to deliver a hydrophobic bioactive compound to its targeted site and exhibit better therapeutic effects [20,21,22].

Polymeric nanoparticles have also shown potentials in dissolving hydrophobic drugs. For example, the development of a vitamin E TPGS/Poloxamer 407 polymeric micelle to increase the solubility of doxorubicin has been found to exhibit better anti-cancer effects compared to free doxorubicin [23]. Patra et al. [24] developed a vitamin E TPGS/Poloxamer micelle that was loaded with quercetin to treat multidrug-resistant breast cancer. Poloxamer 407 (Pol) and D-α-tocopheryl polyethylene glycol 1000 succinate (TPGS) enhanced the solubility and bioavailability of poorly water soluble compounds through encapsulating the compound and loading it into the hydrophobic cargo [23,24]. Furthermore, TPGS could reverse MDR through binding to the ATP of P-gp and reducing drug efflux [25,26,27]. The combination of Lut, Pol, and TPGS may have synergistic effects towards cancer cells, hence reversing MDR. It can be seen to be a perfect alternative and a solution for MDR BC treatment. The aim of this study is to optimize, develop, and characterize the novel drug delivery system to ensure that the micelle has a good EE and size for the efficacy and specificity of the passive targeting of the drug towards cancer cells.

## 2. Materials and Methods

### 2.1. Materials

Luteolin and vitamin E TPGS (tocofersolan) was purchased from MedChem Express (Monmouth Junction, NJ, USA), Poloxamer 407 was purchased from Sigma-Aldrich (St. Louis, MO, USA) and ethanol was purchased from R&M Chemicals Sdn Bhd (Subang, Malaysia).

### 2.2. Preparation of Luteolin-Loaded Micelle

Lut-loaded micelle was prepared using thin film hydration method as described by Patra et al. [24]. Lut was mixed together with TPGS and Pol using ethanol as the solvent until the solution became homogenous. The solvent was then evaporated using a rotary vacuum evaporator and further dried under vacuum overnight until a thin film was produced. About 10 mL of water was used to hydrate the thin film while it was being stirred and heated until a micellar solution was produced. The micellar solution was then centrifuged for 30 min at 5000 rpm and further filtered using a 0.22 µm syringe filter. The final micellar solution was then stored in a deep freezer before being freeze-dried to produce a solid micellar cake powder. The solid micelle was then stored at 4 °C before being used for analysis.

### 2.3. Optimization of Hydration Temperature and Duration

The Lut-loaded micelle was prepared as described in Section 2.2. The heating temperature of the water that hydrated the thin film was manipulated to 10 °C, 25 °C, and 40 °C with constant stirring for 1 h. The micellar solution was then centrifuged and filtered before being evaluated for its EE and PS.

For hydration duration, the micellar solution was hydrated at 25 °C with constant stirring for 0.5 h, 1 h, and 2 h. The micellar solution was centrifuged and filtered before being evaluated for its EE and PS.

### 2.4. Optimization of Freeze-Drying Temperature

A micelle was prepared similarly as described in Section 2.2. After the micelle was hydrated, centrifuged and filtered, the final micellar solution was stored in −20, −50, and −80 °C freezers for 3 h before proceeding to freeze-drying. The freeze-dried micelle was then stored at −4 °C before being evaluated for its EE and PS.

### 2.5. Optimization of Concentration of Copolymers and the Ratio of TPGS:Pol

A Lut-loaded micelle was prepared as described in Section 2.2. The amount of Lut was kept constant while the concentrations of both copolymers were manipulated from 7.5% to 10% and 12.5% (*w*/*v*). The weight ratio of TPGS/Pol was manipulated to 4:0, 3:1, 1:3, and 0:4. The micellar solution of each of the samples was freeze-dried and stored at −4 °C before being evaluated for its EE. The optimized micelles in terms of EE were used for PS analysis.

### 2.6. Encapsulation Efficiency (EE)

This method was adopted from Patra et al. [24] with some modification. Approximately 1 mg of the micellar powder was weighed and added into 5 mL of ethanol and vortexed for 10 s to disrupt the micelle and burst out the encapsulated Lut. The solution was then diluted with ethanol, and the absorbance at 350 nm was measured. The EE was calculated based on the weight of Lut in the micellar powder over the initial amount of Lut used to prepare the micelles. All of the measurements were carried out in triplicate, and the data are presented as mean ± standard deviation (SD). The equation to determine the percentage of EE is as follows:$$\%\mathrm{EE}=\frac{\mathrm{Weight}\;\mathrm{of}\;\mathrm{Lut}\;\mathrm{in}\;\mathrm{the}\;\mathrm{micelle}}{\mathrm{Weight}\;\mathrm{of}\;\mathrm{initial}\;\mathrm{amount}\;\mathrm{of}\;\mathrm{Lut}}\times 100$$

### 2.7. Particle Size Determination and Zeta Potential

The average size of the micelle was determined using the dynamic light scattering method (DLS) (Litesizer 100, Anton Paar, Graz, Austria). The cell temperature was 25 °C with a detection angle of 175°. The micellar cake was dissolved up to 1 mg/mL and diluted 10× before filtering using a 0.22 µL syringe filter and was then put into the cuvette.

The zeta potential of the micelle was determined using a zetasizer (Malvern Instrument, Malvern, UK) at 25 °C. The micellar cake of the optimized micelle was dissolved up to 1 mg/mL and diluted 10× before filtering using a 0.22 µL syringe filter and put into the cuvette.

All of the measurements were performed in triplicate after dilution using filtered distilled water, and the data are presented as mean ± SD.

### 2.8. Transmission Electron Microscopy (TEM)

Transmission electron microscopy (TEM) (Libra 120, Carl Zeiss, Jena, Germany) was used in this study to validate the size obtained from DLS and determine the surface morphology. A drop of micellar solution was placed on a copper grid and stained with phosphotungstic acid solution (2%, *w*/*v*) for 30 s. After the excess solution was removed, the sample was dried in air and the morphology of the micelle was observed under TEM.

### 2.9. Solubility Study

This method was adopted from Patra et al. [24] with some modification. An excess amount of Lut powder was dissolved in distilled water through stirring at 120 rpm and at 25 °C for 72 h using an incubator shaker followed by centrifugation at 5000 rpm for 30 min. The supernatant was filtered using a 0.22 µm syringe filter. The filtered solution was then analyzed for its concentration using UV spectrophotometer (Uviline 9400, Secomam, Mainz, Germany) at 350 nm. To determine the solubility of Lut in micellar solution, 1 mL of the micellar solution obtained after hydration, centrifugation, and filtration was added into 5 mL of ethanol and vortexed to disrupt the micelle and burst out the encapsulated Lut. The solution was then diluted with ethanol, and the absorbance at 350 nm was measured.

### 2.10. CMC Determination

The critical micelle concentration (CMC) of the TPGS/Pol micelle was determined using iodine as a hydrophobic probe [24,28]. About 0.5 g of iodine and 1 g of potassium iodide were weighed and dissolved together in 50 mL distilled water as KI/I2 standard solution. The blank optimized micelle solution was prepared into a series of dilutions with different concentrations ranging from 0.1% to 0.000001% of polymer concentration. About 200 µL of KI/I2 standard solution was added into each of the dilution series. The mixtures were incubated in a dark room at room temperature for 12 h before the absorbance was measured at 366 nm using a UV-Vis spectrophotometer (Uviline 9400, Secomam, Mainz, Germany). The curve of absorbance against log polymer concentration was plotted to determine CMC, where the value of CMC corresponds to the polymer concentration when a rapid increase in the absorbance was observed.

### 2.11. FTIR Study of Micelle

The interaction between Lut and the copolymers in the micelle was characterized using Fourier-transform infrared spectroscopy (FTIR). Pure Lut, pure TPGS, pure Pol, and freeze-dried Lut-loaded micelle were taken out for FTIR study. The FTIR spectra were recorded with a FTIR spectrophotometer using the potassium bromide (KBr) disk method. The FTIR spectra were scanned in the IR range from 400 to 4000 cm^−1^.

### 2.12. Crystallinity Study of Micelle

To investigate the crystallinity of freeze-dried Lut-loaded micelle powder, X-ray diffraction (XRD) analysis was performed on the pure Lut, pure TPGS, pure Pol, blank lyophilized micelle, and lyophilised Lut-loaded micelle. The X-ray diffraction patterns were obtained using a X-ray diffractometer.

### 2.13. In vitro Drug Release

The release behavior of the Lut-loaded micelle was investigated and compared to free Lut using the dialysis method [24,29]. About 1 mg equivalent weight of Lut-loaded micelle and 1 mg of Lut in ethanol were separately suspended in a dialysis membrane bag, with both ends sealed, and were immersed in 100 mL release media consisting of phosphate-buffered saline (PBS; pH 7.4) solution with 0.5% Tween 80 at 37 °C under horizontal shaking (100 rpm/min). At predetermined period, 1ml aliquot of the release media was taken to measure the absorbance at 350 nm using UV-Vis spectrophotometer. About 1ml of fresh release media was replaced each time the aliquot of the release media was taken for absorbance measurement. The release behavior of Lut-loaded micelle was also measured in release media with pH 6.8 to mimic the microenvironment of tumor cells since the pH of intracellular of tumor cells is 6.7–7.1 [30,31].

Mathematical models were applied on the data obtained from the release study of the micelle at different pH to predict the release profile. Those models included the zero-order model, first-order model, Kosmeyer–Peppas model, Hixson–Crowell model, and Higuchi model, with the following equations:$$\mathrm{Zero-order}\;\mathrm{model}:\;\frac{M_{t}}{M_{\infty }}=k_{0}t$$
$$\mathrm{First-order}\;\mathrm{model}:\;\ln\left(1-\frac{M_{t}}{M_{\infty }}\right)=-k_{1}t$$
$$\mathrm{Kosmeyer}–\mathrm{Peppas}\;\mathrm{model}:\;\frac{M_{t}}{M_{\infty }}=k_{kp}t^{n}$$
$$\mathrm{Hixson}–\mathrm{Crowell}\;\mathrm{model}:\;1-{\left(1-\frac{M_{t}}{M_{\infty }}\right)}^{\frac{1}{3}}=1-k_{HC}^{t}$$
$$\mathrm{Higuchi}\;\mathrm{model}:\;\frac{M_{t}}{M_{\infty }}=k_{H}t^{\frac{1}{2}}$$
where *M_t_* is the cumulative amount of drug release at time point t; *M_∞_* is the initial amount of the drug; *t* represent time; *k*_0_, *k*_1_, *k_KP_*, *k_HC_*, *k_H_* represent the zero-order, first-order, Kosmeyer–Peppas, Hixson–Crowell, and Higuchi rate constants, respectively; and *n* represents the release exponential.

### 2.14. Statistical Analysis

Data are presented as mean ± SD. One-way analysis of variance (ANOVA) followed by Tukey’s post hoc multiple comparisons for significance were used throughout the analysis of the data using JASP SOFTWARE (0.16.1). *p* < 0.05 was considered statistically significant.

## 3. Results and Discussion

### 3.1. Effect of Polymer Concentration and Polymer Ratio on EE and PS

The range of EE obtained was 77.0–92.3% as shown in Table 1. The EE of the micelle with a polymer concentration of 10 is significantly higher than that of 7.5 (*p* < 0.05). This indicates that the higher the concentration of polymers, the higher the EE of the micelles. The higher concentration of polymer increases the capacity of the micellar core, resulting in the ability to load a more hydrophobic compound [32,33]. However, EEs of 10% and 12.5% have no significant difference, even though 12.5% was slightly higher, indicating that the EE–concentration relationship has reached its plateau at the concentration of 10–12.5%. The EE of the TPGS micelle (ratio 4:0) was significantly higher than the Pol micelle (0:4). This can be due to the high hydrophobicity of TPGS. The higher the amount of hydrophobic core, the higher the amount of drug can be loaded into the micelle. The ratio 3:1 is significantly higher than the other ratio. These results were in accordance to the study conducted by Fares et al. [32], where the increase in hydrophobic ratio increased EE significantly. Therefore, the optimum ratio for the optimized micelle is 3:1.

Nanoparticle encapsulation efficiency refers to the ability of nanoparticles to effectively encapsulate and retain a desired substance, such as drugs, proteins, or other bioactive compounds, within their structure. It is an important parameter that determines the effectiveness and stability of nanoparticle-based delivery systems. One of a few key factors that affect the encapsulation efficiency is the choice of materials in the formulation of nanoparticles. The EE of the mixed polymeric micelle in this study is much higher if compared to its individual polymeric micelle, as shown in Table 1. Chang et al. [34] demonstrated that the EE of a curcumin-loaded micelle is higher in mixed micelles composed of PEGMEMA 12:PS 595, where it has higher encapsulation efficiency compared to single micelles composed of PEO–PCL. Understanding and optimizing encapsulation efficiency is essential for the successful design and application of nanoparticle-based delivery systems.

Since the EE of 10(3:1) and 12(3:1) is significantly higher than the other formulations, these two formulations can be considered as optimized micelles. However, due to the insignificant difference between these two formulations in terms of EE, we have chosen 10(3:1) for the next characterization studies. Akbar et al. [35] suggested that higher EE can be a potential candidate for delivering active compounds as it can enhance the bioavailability of the compound. Other factors to take into consideration in selecting an optimized micelle are cost and reproducibility. Micelle 10(3:1) used a lesser amount of polymers; therefore, more micelles can be produced for other tests.

The PS of the optimized micelle is compared to other micelles with the same concentration but different ratios [10(4:0), 10(1:3) and 10(0:4)] and micelles with the same ratio but different concentrations [12.5(3:1) and 7.5(3:1)], as shown in Table 2 and Table 3, respectively. The range of PSs for all the micelles analyzed was 18.18 nm–28.65 nm.

The PS of the 4:0 ratio micelle was significantly higher than that of 0:4, which indicates that TPGS produced a smaller micelle than Pol. The incorporation of both polymers produced a bigger-size micelle than its individual polymers.

This can be attributed to the hydrophilicity of Pol. Wei et al. and Fares et al. [32,36] agreed that the addition of a hydrophilic head from F127 can be the reason why the micellar size became bigger, and the low amount of hydrophilic polymer might reduce the size of the micelle. There was also a significant difference between the PS of 10(3:1) and 12.5(3:1), which makes it clearer that 10(3:1) is the optimized micelle and was chosen for the next tests and characterizations.

The PS obtained via DLS has been confirmed with a transmission electron microscope (TEM) as shown in Figure 1. In Figure 1B, the micelle formed was spherical in shape and had two layers of color, light grey and black. The grey layer indicates the hydrophilic region while black layer indicates the hydrophobic region. We predicted that Lut was dissolved and encapsulated in the hydrophobic region of the micelle.

In addition, the PS of the blank optimized micelle was measured in comparison to the optimized micelle. The PS of the blank optimized micelle was insignificantly lower than the optimized micelle, which was 26.97 ± 1.11 nm. The increase in PS between the Lut-loaded micelle and the blank micelle might be due to the loading of Lut into the micelle. This result was comparable with the study conducted by Basir et al. [37], where the blank TPGS-PEG micelle has a lower PS compared to the micelle that was loaded with naringenin and gallic acid.

Nanoparticle size is a crucial characteristic that significantly impacts the properties and behavior of nanoparticles. The size of nanoparticles can range from 1 to 1000 nm [38]. PS is vital to be monitored in nanoparticle studies due to its objective to penetrate tumorous cells via the EPR effect. A micelle can reside in a tumor’s blood vessel if the particle size is less than 200 nm. Several studies have stated that the range of size of nanoparticles that can benefit the EPR effect is between 1–400 nm [39]. However, if the size of the micelle is higher than 200 nm, the micelle might be eliminated from the body via RES [39,40]. Therefore, it is important to monitor the size of the nanoparticles to ensure high efficacy of the drug carried to the site of tumor. The PSs between nanoparticles also differ depending on the types and systems of nanoparticles. For example, the polymeric micelle has been found mostly to have a PS of <80 nm [26,32]. Liposome has been reported to have a PS less than 200 nm [20,41], while Niosome has been reported to have a PS less than 600 nm [21,42]. It is important to note that these are just a few examples from the provided references, and the particle size ranges can vary significantly depending on the specific study and nanoparticle system. The size range of nanoparticles can have a significant impact on their properties and applications. Smaller nanoparticles often exhibit different magnetic, optical, and catalytic properties compared to larger nanoparticles. Additionally, the size of nanoparticles can influence their behavior in terms of aggregation, transport, and cellular uptake. Therefore, controlling and characterizing the particle size is crucial for understanding and optimizing the performance of nanoparticles.

After confirmation of the optimized micelle, the micelle was then tested for its zeta potential to determine the tendency of the micelle to aggregate due to the charge carried by the micelle. The surface charge of a nanoparticle is indicated by its zeta potential. It characterizes the electric potential of nanoparticles and is influenced by both the particle composition and the dispersing medium. According to Raval et al. [43], nanoparticles exhibiting zeta potentials exceeding +30 mV or falling below −30 mV are regarded as a stable colloidal suspension system, effectively preventing nanoparticle aggregation. Conversely, nanoparticles with zeta potential values ranging from +30 mV to −30 mV indicate inadequate colloidal stability and are prone to flocculation, agglomeration, or aggregation. A dispersion or suspension featuring a low zeta potential value enhances nanoparticle aggregation due to the influence of van der Waals attractions.

The zeta potential of the optimized micelle was −30.97 ± 1.93 mV. Previous studies have shown that the zeta potential of a TPGS/Pol micelle has much lower zeta potential and has the tendency to become unstable due to aggregation. Shen et al. [26] developed a TPGS/Pol micelle to encapsulate glycyrrhizic acid with a zeta potential of −5.92 ± 0.68 mV. Grimaudo et al. [44] encapsulated cyclosporine with a TPGS/Pol micelle and obtained a zeta potential of −4.040 ± 3.04 mV. In this study, the zeta potential was higher, so it can be assumed that the micelle we developed possessed high stability for the long term.

### 3.2. The Effect of Hydration Temperature and Duration on EE and PS

The film hydration method has been widely used in the formation of micelles. This is due to the fact that this method is easy, highly producible, and cost-effective, compared to other methods of forming micelles [45]. In most studies, the film hydration method starts via stirring the mixture of the polymers and poorly soluble drug inside a solvent that can dissolve these materials and is highly volatile. The solvent is then evaporated, leaving behind a homogenous mixture of polymers and drug in the form of a thin film which is then hydrated using water or buffer solution. The poorly soluble drug becomes soluble in aqueous solution due to the entrapment of the drug into the hydrophilic region of the micelle that is formed by the polymer.

The hydration temperatures of the thin film of micelles were manipulated to 10 °C, 25 °C, and 40 °C, and the EE and PS were monitored in this study. From Table 4, it can be seen that when the temperature was low, the EE became lower and the PS became bigger. As the temperature rose to 25 °C, the EE became higher and the PS became smaller. However, when the temperature was increased further, the EE became lower and the PS became smaller. Therefore, the optimized hydration temperature for the micelle was found to be 25 °C.

From the results in Table 4, it can be seen that the optimized temperature for hydration that produced high EE and lower PS was 25 °C. The probable reason is because when increasing the temperature from 10 °C to 25 °C, the intermolecular kinetic energy increases; therefore, increasing the amount of hydrophobic bond between the drug and the hydrophobic core of the micelle produces a more compact hydrophobic core, thus decreasing in micellar size and increasing the amount of drug loaded. Increasing the temperature from 25 °C to 40 °C might alter the acyl chain of the micellar core, causing the release of Lut. This might be due to the break of hydrophobic and hydrogen bonds between the acyl group of the micellar core, thus making the micellar core less compact and appear bigger [46]. Furthermore, increasing the temperature of hydration above 25 °C can lead to the dehydration of Pol PEO heads and more interaction in the mixed polymeric micelle, causing more aggregation [47]. Therefore, hydrating the micellar film at 25 °C might yield the most optimum micelle in terms of EE and PS.

For hydration duration, the EE and PS were observed when the duration of the hydration of the micellar thin film was manipulated to 0.5 h, 1 h, and 2 h. Table 5 shows that the higher the duration of hydration, the higher the EE, while the PS becomes smaller. However, when the duration of the hydration was prolonged for a longer period, the EE was reduced and the PS became bigger. Therefore, the optimized hydration temperature and duration is 25 °C for 1 h.

From the results in Table 5, the longer the duration of hydration, the higher the EE and PS. This might be due to the increase in the partitioning of Lut in self-assembled micelles [47]. Furthermore, a shorter duration of hydration might cause the thin film to not fully dissolve in the hydration solvent, causing a lesser amount of Lut to be present in the micellar solution. Ai et al. [45] demonstrated an increase in the EE of doxorubicin in a polyethylene glycol 5000-lysine-di-tocopherol succinate micelle when the hydration duration was increased. Increasing the hydration duration might also reduce the PS of the micelle due to more compact packing of the micellar core with an increase in hydrophobic Lut content. Nasehi et al. [47] agreed that increasing the hydration duration might increase the EE and reduce the PS of sofarenib-loaded micelles. Fattahi et al. [48] stated that a more tightly packed micelle would be formed if the hydrophobicity of the polymer increased. Therefore, it is best to hydrate the film for 1 h instead of 30 min.

### 3.3. The Effect of Freeze-Drying Temperature on EE and PS

Nanoparticles, e.g., polymeric micelles, have to face the challenge of instability since they are produced in the form of a solution. The instability of nanoparticles may come from physical (aggregation and drug leakage) and chemical instability (oxidation and hydrolysis) [49]. One of the methods to ensure the stability of nanoparticles is to immobilize the particle through changing its solution state to a solid state via freeze-drying or lyophilization. While the freeze-drying process itself can lead to the aggregation of nanoparticles through freezing, it is important to make sure that the freezing temperature is monitored and optimized so that the nanoparticles produced are stable, maintained in nano-size, and efficacious.

In this study, the micellar solution produced after hydration of a thin film with water was centrifuged and filtered to filter out the unincorporated Lut. The clear micellar solution was then deep-frozen before freeze-drying. The temperature of the freezing temperature was manipulated from −20 °C to −50 °C and −80 °C, and the cakes of micelles produced after freeze-drying were analyzed for their EE and PS. In Table 6, it can be seen that the EE of the micelle that was frozen in the −80 °C freezer has a significantly higher amount compared to those that were frozen in the −20 °C and −50 °C freezers. The same has been observed in terms of PS, where the micelle that was frozen at −80 °C was significantly smaller than those frozen at −20 °C and −50 °C. However, there was no significant difference between −20 °C and −50 °C in terms of EE and PS, which indicates that micellar solution should be frozen in a −80 °C freezer before freeze-drying to yield an optimum micelle.

In the process of freeze-drying, nanoparticles would need to undergo three steps: (1) freezing, (2) primary drying, and (3) secondary drying. Freezing is vital for nanoparticle stability since it could immobilize the movement of colloid particles via Brownian motion. When freezing, the nanoparticles will be separated into multiple phases, including a crystal ice phase from frozen aqueous solution, and the solute, which is the nanoparticle itself. The freezing temperature plays a part in determining the efficiency of lyophilization. If the freezing temperature is higher than the glass transition temperature (Tg’) of the drug, then the drug will most likely not enter a fully frozen state, which will affect the final product of lyophilization exhibiting issues such as drug leakage, which can affect the EE of the micelle [50,51,52]. This might be the reason why the EE of the Lut-loaded micelle in this study showed a higher EE when the micelle was frozen in −80 °C compared to other temperatures. The Tg’ of the micelle might be between −78 °C and −52 °C, as Tang and Pikal [52] suggested that a temperature of at least −2 °C below Tg’ is required for complete freezing. The incomplete freezing of the micelle might also cause the PS of the micelle to appear bigger, as suggested in this study. Moretton et al. [53] shared the same opinion, as a rifampicin-loaded micelle appeared bigger when frozen in a lower-temperature freezer compared to a higher-temperature freezer. There were not many studies that highlight the importance of the freeze-drying temperature on the EE and PS of micelles. Therefore, further studies are needed to have a more thorough explanation regarding this experiment.

### 3.4. Solubility Study

Most anticancer drugs, such as paclitaxel and docetaxel, have exhibited low solubility in water. The same goes for compounds that possess high potential anticancer properties such as quercetin and curcumin. The average range of solubility of these potential and developed anticancer compounds are in the µg/mL range [39,54]. Nanoparticles have shown great potential in delivering hydrophobic drugs, which are characterized by poor water solubility. Various types of nanoparticles have been explored for the delivery of hydrophobic drugs. For example, polymeric micelles encapsulate quercetin to treat breast cancer [24], solid-lipid nanoparticles are used for the delivery of essential oils into the target site of cancer cells [22], and nanoliposome has been utilized to deliver phenolic compounds to colorectal cancer cells in a mouse model [20].

The solubility of Lut in water is 30.67 µg/mL, while the solubility of luteolin when encapsulated with a TPGS/Pol micelle is 2594.02 µg/mL. The solubility of Lut in a micelle is 459-fold more soluble when compared to pure Lut in water. This indicates that the TPGS/Pol micelle can be used to increase the solubility of hydrophobic drugs such as Lut in water.

The effectiveness of nanoparticles as solubilizers for hydrophobic drugs was discussed in a study conducted by Grimaudo et al. [44], where they encapsulated hydrophobic cyclosporine using a TPGS/Pol micelle for corneal use. The solubility of cyclosporine loaded into the TPGS/Pol micelle enhanced tremendously, about 107-fold compared to pure cyclosporine in water. Gadadare et al. [55] demonstrated that the solubility of repaglinide increased up to 25-fold compared to the free drug when TPGS was used as its component for nanocrystals. Therefore, the utilization of micelles in delivering hydrophobic drugs can be very useful as the encapsulation of hydrophobic drugs into micelles has been proven to increase the solubility of Lut.

### 3.5. CMC Determination

Critical micelle concentration is the minimum concentration of the surfactant needed to self-assemble and encapsulate to become a micelle. Below the CMC level, the molecules of the surfactant line up at the surface of the water, with the hydrophobic region facing upward and away from water while the hydrophilic region faces downward and is in contact with water molecule. When the concentration of the surfactant exceeds the CMC level, the molecule of the surfactant will self-assemble, during which the hydrophilic region will encapsulate, and the hydrophobic region will be at the core of the micelle.

In this study, the CMC of the obtained TPGS/Pol micelle was 0.001% *w*/*v*, as shown in Figure 2. This result was agreeable with other studies that found the CMC of TPGS/Pol micelles was at 0.0013% *w*/*v* [28] and 0.0015% *w*/*v* [24]. According to literature, the CMC of the TPGS micelle and Pol micelle are 0.00052% *w*/*v* and 0.0575% *w*/*v* [44]. The mixture of TPGS and Pol in forming micelles causes the CMC to have an intermediate value between pure TPGS and Pol micelles. Furthermore, the CMC value of the mixed TPGS/Pol micelle was shifted towards pure TPGS micelle’s CMC value. This is because the amount of TPGS is higher than Pol in the composition of the optimized mixed micelle [24,32,44]. The determination of CMC is very important in the study of nanoparticles, especially micelles. This is due to the fact that the micelle can be disassembled when the micelle undergoes extreme dilution below CMC level in body fluid. When this happens, the purpose of transporting drugs into the specific site of action cannot be achieved. Therefore, micelles with lower CMC have higher survivability and stability in body fluids and can transport the drugs effectively to the site of action.

### 3.6. FTIR

To investigate the interaction between the drug and the polymers in the micelle produced, an FTIR study was carried out. The spectra observed were pure Lut, TPGS, Pol, and the Lut-loaded TPGS/Pol micelle, as shown in Figure 3. The pure Lut sample showed the main characteristic bonds at 3418 cm^−1^ (strong -OH stretching), 1653 cm^−1^ (medium C=C alkene stretching), and 1167 cm^−1^ (strong C-O-C stretch). The pure TPGS showed peaks such as strong C=O ester stretching at 1736 cm^−1^. Pure Pol showed peaks at 3448 cm^−1^ (strong O-H stretching). Both TPGS and Pol shared the same peaks at 2884 cm^−1^ (medium C-H alkane stretching), 1465 cm^−1^ (medium C-H methylene bending), and 1344 cm^−1^ (medium O-H alcohol bending). In the spectrum of Lut-loaded micelle sample, Lut absorption bonds can be seen with no difference in the locations of the absorption bands, indicating that there was no interaction between the drug and the polymers. The spectrum obtained can be validated by other studies that also obtained the same range of spectra [56,57,58].

### 3.7. Crystallinity Study Using XRD

The X-ray diffractometry technique is very useful to characterize the crystal and crystallographic phase which determines the physical properties of nanoparticles. It is a non-destructive technique that it has the ability to gather the information of the average of the particles, unlike direct imaging techniques, e.g., electron microscopy, where only a small sample of particles can be studied, which may not be truly representative of the material [59,60].

In this study, XRD analysis was performed to determine the physical state of Lut encapsulated in micelles as to compare with free Lut. As shown in Figure 4, there were two characteristic Bragg peaks of pure Lut in the 2θ of 9° and 28°. The presence of the peaks indicates that the physical state of Lut was crystalline in structure. On the other hand, there were no characteristic Bragg peaks of Lut seen in the Lut-loaded micelle with decreased crystallinity, which might be due to the drug already being molecularly dispersed and entrapped in the amorphous state of the micelle. Both polymers shared the same peak at 2θ = 17° and 23°, with Pol showing a more intense peak than TPGS, indicating that Pol has a higher crystallinity than TPGS. This result aligned with a previous study that reported the same peak for TPGS and Pol micelles [26]. In comparison with the blank micelle, both peaks were still present but with lower intensity, suggesting that the mixture of both polymers decreased their crystallinity so that they became amorphous-state micelles. There was no variation in the blank TPGS/Pol micelle peaks compared to the Lut-loaded micelle, which could suggest that there was no interaction between Lut and the polymers.

This result is aligned with a study of a paclitaxel-loaded chitosan micelle, where Liang et al. [61] found that the intensity of the peak of paclitaxel became non-existent when paclitaxel was encapsulated in the polymeric micelle. This indicates that paclitaxel changed its crystallinity from crystal to amorphous when encapsulated in a polymeric micelle. Gupta et al. [62] also shared the same opinion; the authors suggested that curcumin was dispersed in an amorphous state when entrapped in the micelle. These studies suggested that Lut was encapsulated in polymeric micelles in a molecular or amorphous state. It is also a clear indication that the solubility of Lut increased as Lut transited from crystalline to amorphous. Eerdubrugh et al. [63] stated that the increase in the drugs’ solubility is the result of the higher energy state of the material due to the nanosizing process, which arises from partial amorphization. Therefore, the solubility of Lut is also affected by its own crystalline state.

### 3.8. In Vitro Drug Release Study

The in vitro release behavior of Lut was investigated using the dialysis method, with PBS (pH 7.4) and 0.5% Tween 80 used as release media to receive the sink condition. The release of Lut without micelles was found to be rapid and reach 100% in less than 4 h. On the other hand, a different trend was observed in the release of Lut that was loaded in micelles. There was an initial rapid release observed for the first 10 h of the study for Lut-loaded micelles in both media, in which the release of Lut into the media steadily increased over the hour. Lut was released steadily for up to 7 days when loaded into micelles at physiological pH (pH 7.4). However, the release of Lut in pH 6.8 was observed to be higher than in pH 7.4. This might due to the partitioning of Lut in acidic environments, which makes Lut more soluble in lower pH [30]. With this information, it is useful to know that Lut can be released in high amounts in slightly acidic tumor cells but survives longer in body fluid with physiological pH.

As shown in Figure 5, the release of free Lut was more rapid compared to the release of Lut-loaded micelles, which was more sustained and can last up to 7 days. This finding agrees with previous studies that showed the burst-like release of Lut without micelles and the sustained release of Lut when loaded into micelles [16,29,64,65,66]. The sustained release behavior that was observed in this study may be caused by several factors: (1) the diffusion of Lut from the micelle to the release medium; (2) the degradation and hydrolysis of the polymeric micelle, causing Lut to be released out of the micelle; and (3) polymer erosion and swelling [16,24].

The release profile of the Lut-loaded micelle at different pH and free Lut in ethanol at pH 7.4 was fitted into mathematical models to elucidate the mechanism and kinetics of drug release, as shown in Table 7. According to the R^2^ value of these various models, the Lut-loaded micelle release profile fit best to the Kosmeyer–Peppas model (pH 7.4: 0.9611; pH 6.8: 0.9760), whereas the free Lut fit the first-order model (R^2^: 0.9559) the best. The value of ‘*n*’ denotes various mechanisms for the release of the drug from the carriers. According to Shen et al. [26], for carriers like micelles with an aspect ratio (diameter/length) in the order of 1, *n* < 0.43 corresponds to Fickian diffusion (Case I), whereas 0.43 < *n* < 0.85 indicates non-Fickian or anomalous diffusion, and *n* > 0.85 indicates non-Fickian Case II release kinetics. The release of Lut-loaded micelles in pH 7.4 is in accordance with non-Fickian Case II release (*n* > 0.85), which indicates that the mechanism driving the drug release is the swelling or relaxation of the polymeric chain, whereas the release of Lut-loaded micelles at pH 6.8 is in accordance with anomalous non-Fickian diffusion, suggesting that the mechanism of release of Lut at pH 6.8 is a combination of erosion and diffusion of the polymeric matrix.

## 4. Conclusions

A luteolin-loaded micelle of TPGS and poloxamer 407 was prepared and optimized using the film hydration method. The optimized ratio of these two copolymers was found to be 3:1, while the preferred concentration of copolymers used was 10% (*w*/*v*). The hydration temperature and duration used in the film hydration method used were 25 °C and 1 h, respectively. The freezing temperature used before lyophilization was −80 °C. All of these optimized parameters produced higher encapsulation efficiencies and lower particle sizes. The developed micelle can also withstand a higher volume of dilution (up to 0.001 mg/mL) with sustained-release behavior in blood pH and higher solubility in tumor microenvironment pH. This study suggests that this developed mixed micelle can be used to solubilize hydrophobic anticancer compounds. However, the effectiveness of the micelle has not been proven yet. Therefore, the formulation should be tested in vitro and in vivo to confirm the effectiveness of this formulation.

## Figures and Tables

**Figure 1 cancers-15-03741-f001:**
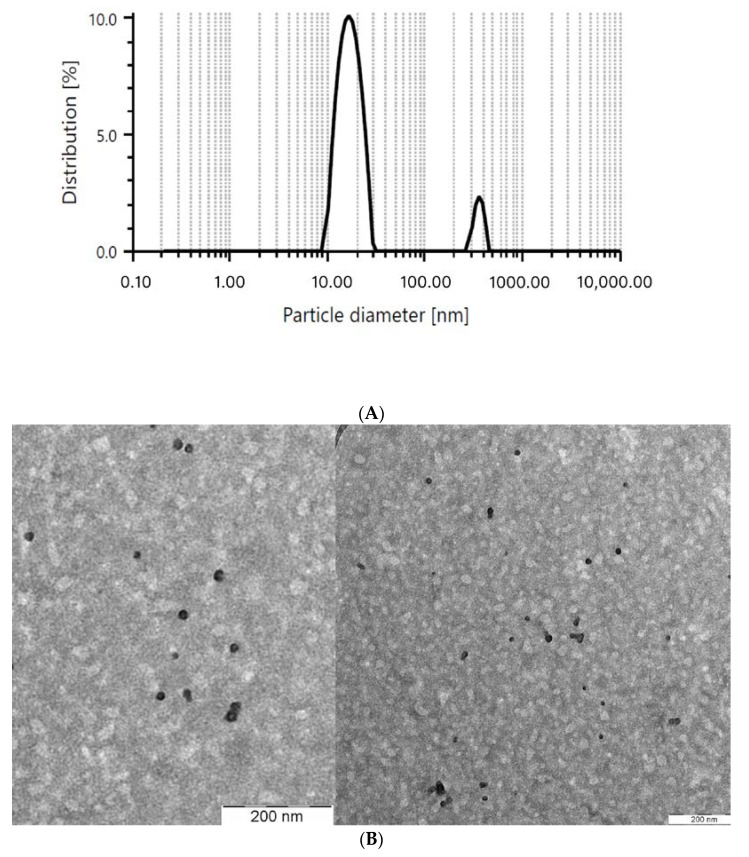
PS of the optimized micelle obtained via (**A**) dynamic light scattering (DLS) and (**B**) transmission electron microscope (TEM).

**Figure 2 cancers-15-03741-f002:**
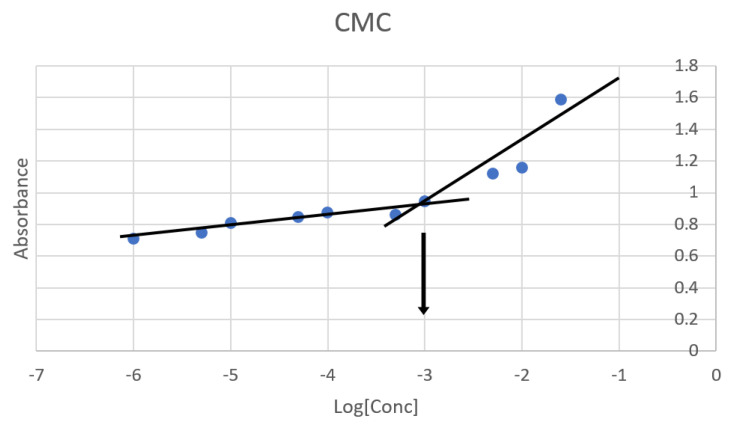
Critical micelle concentration (CMC) of the optimized micelle determined via KI/I_2_ probe. Blue dot is the absorbance at the given concentration. Black lines are linear curves of ‘gradual increase’ (from −6 to −3) in absorbance and ‘rapid increase’ (from −3.3 to −1.6) in absorbance. Black arrow determines the CMC where the two black lines intersected.

**Figure 3 cancers-15-03741-f003:**
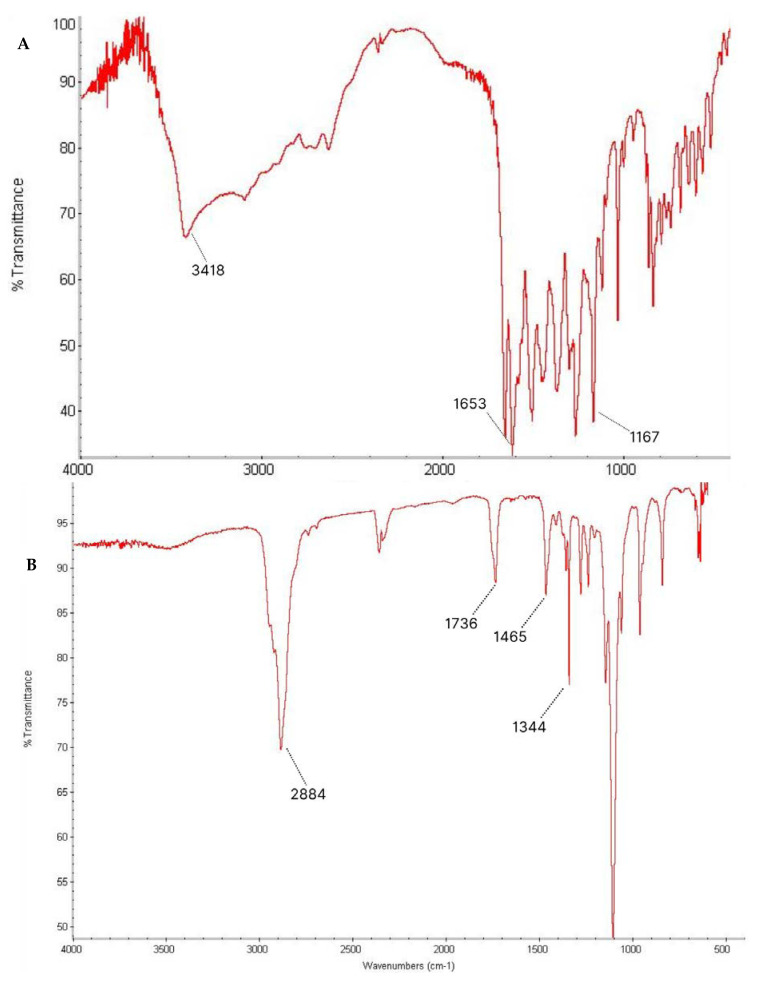
Fourier-transform infrared (FTIR) spectrum of (**A**) Lut, (**B**) TPGS, (**C**) Pol, and (**D**) sample.

**Figure 4 cancers-15-03741-f004:**
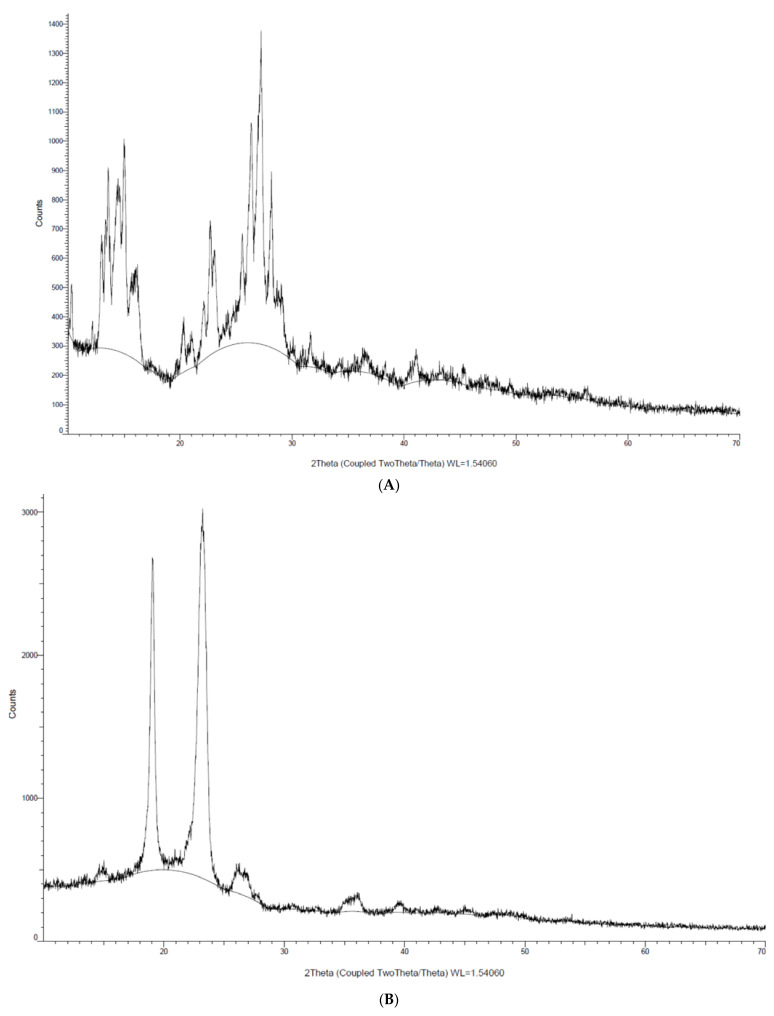
Crystallinity analysis of (**A**) Lut, (**B**) Pol, (**C**) TPGS, (**D**) blank micelle and (**E**) Lut-loaded micelle via XRD.

**Figure 5 cancers-15-03741-f005:**
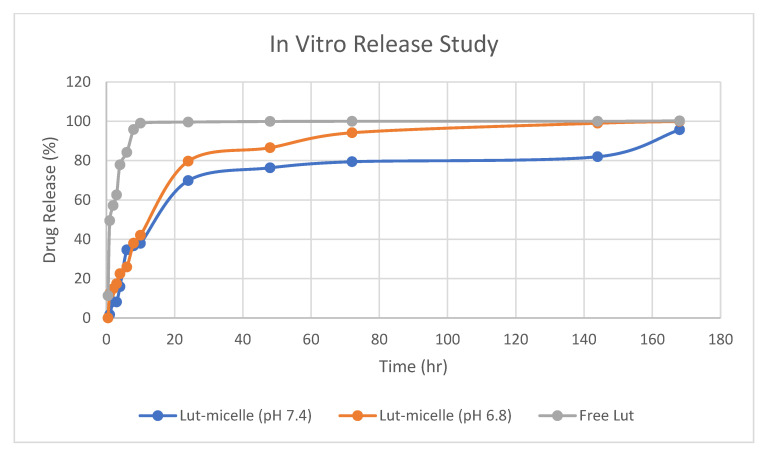
Drug release study of free Lut (grey), Lut-loaded micelle in pH 7.4 (blue), and Lut-loaded micelle in pH 6.8 (orange).

**Table 1 cancers-15-03741-t001:** Encapsulation efficiency (EE) of Lut-loaded TPGS/Pol micelle with different concentrations of co-polymers and ratios of TPGS/Pol (V:P).

Concentration (% *w*/*v*)	7.5	10	12.5
Ratio (V:P)	EE (% *w*/*w*)		
4:0	78.6 ± 0.6	80.3 ± 0.5	86.2 ± 0.5 *
3:1	85.5 ± 1.0 *	90.7 ± 0.9 *	92.3 ± 0.6 *
1:3	77.7 ± 0.2	79.9 ± 0.3	77.4 ± 0.5
0:4	77.0 ± 0.7	78.5 ± 0.4	74.3 ± 0.4

* *p* < 0.05 shows statistical significance.

**Table 2 cancers-15-03741-t002:** Comparison of EE and PS between optimized micelles and other micelles with the same (V:P) ratio but different co-polymer concentrations.

Sample	EE (% *w*/*w*)	PS (nm)
7.5(3:1)	85.5 ± 1.0	26.72 ± 1.55
10(3:1)	90.7 ± 0.9 *	24.57 ± 0.61 *
12.5(3:1)	92.3 ± 0.6	28.65 ± 1.32 *

* *p* < 0.05 shows statistical significance.

**Table 3 cancers-15-03741-t003:** Comparison of EE and PS between optimized micelles and other micelles with the same co-polymer concentration but different (V:P) ratios.

Sample	EE (% *w*/*w*)	PS (nm)
10(4:0)	80.3 ± 0.5	18.18 ± 1.01 *
10(3:1)	90.7 ± 0.9 *	24.57 ± 0.61 *
10(1:3)	79.9 ± 0.3	27.65 ± 1.11 *
10(0:4)	78.5 ± 0.4	22.56 ± 0.66

* *p* < 0.05 shows statistical significance.

**Table 4 cancers-15-03741-t004:** The effect of hydration temperature of the optimized micelle on its EE and PS.

Temperature	EE (% *w*/*w*)	PS (nm)
10 °C	80.1 ± 1.2	26.90 ± 2.23
25 °C	90.7 ± 0.9 *	19.97 ± 2.21 *
40 °C	83.0 ± 1.0	25.72 ± 1.27

* *p* < 0.05 shows statistical significance.

**Table 5 cancers-15-03741-t005:** The effect of hydration duration of the optimized micelle on its EE and PS.

Duration (h)	EE (% *w*/*w*)	PS (nm)
0.5	92.7 ± 0.3	26.27 ± 3.35
1	95.2 ± 0.2 *	19.97 ± 2.23 *
2	89.2 ± 0.3	23.43 ± 0.91

* *p* < 0.05 shows statistical significance.

**Table 6 cancers-15-03741-t006:** Effect of the freezing temperature of the optimized micelles on its EE and PS.

Temperature	EE (%)	PS (nm)
−20 °C	84.0 ± 0.5	37.40 ± 1.21
−50 °C	83.1 ± 0.4	36.32 ± 1.46
−80 °C	86.6 ± 0.3 *	28.65 ± 1.22 *

* *p* < 0.05 shows statistical significance.

**Table 7 cancers-15-03741-t007:** Drug release profile of Lut according to mathematical models.

Mathematical Models		pH 7.4 ± SD	pH 6.8 ± SD	Free Lut (pH 7.4) ± SD
Zero order	k	0.6972 ± 0.01	0.7851 ± 0.02	0.8479 ± 0.02
	R^2^	0.8204 ± 0.02	0.8267 ± 0.02	0.5018 ± 0.01
First order	k	0.0372 ± 0.005	0.0383 ± 0.002	0.8148 ± 0.001
	R^2^	0.5996 ± 0.001	0.5971 ± 0.002	0.9559 ± 0.001
Kosmeyer–Peppas	k	4.2778 ± 0.09	8.2065 ± 0.1	36.4900 ± 0.08
	*n*	1.001 ± 0.1	0.7106 ± 0.09	0.5587 ± 0.05
	R^2^	0.9611 ± 0.01	0.9760 ± 0.01	0.875 ± 0.01
Hixson Crowell	k	0.0084 ± 0.001	0.0091 ± 0.001	0.0100 ± 0.002
	R^2^	0.9553 ± 0.001	0.9705 ±0.002	0.6683 ± 0.002
Higuchi	k	8.4327 ± 0.2	9.5504 ± 0.1	11.6609 ± 0.5
	R^2^	0.9219 ± 0.002	0.9292 ± 0.002	0.6452 ± 0.03

## Data Availability

The data presented in this study are available on request from the corresponding authors.

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
