# Peer review of "Optimization of a Luteolin-Loaded TPGS/Poloxamer 407 Nanomicelle: The Effects of Copolymers, Hydration Temperature and Duration, and Freezing Temperature on Encapsulation Efficiency, Particle Size, and Solubility"

_cancers, 2023, doi:10.3390/cancers15143741_

Round 1
Reviewer 1 Report
The authors submitted the MS “Optimization of Luteolin-Loaded TPGS/Poloxamer 407 Nano micelle: Effect of Copolymers, Hydration Temperature and Duration, and Freezing Temperature on Encapsulation Efficiency, Particle Size, and Solubility.
Muhammad Redza Fahmi Mod Razif, Siok Yee Chan, Riyanto Teguh Widodo, Yik-Ling Chew, Masriana Hassan, Shairyzah Ahmad Hisham, Shamima Abdul Rahman, Ching Siang Tan, Siew-Keah Lee and Kai Bin Liew” for possible publication at the Journal.
In Introduction described that “Breast cancer (BC) is the most diagnosed cancer and causing about 0.7 million deaths worldwide in 2020. BC is mostly treated with surgery, chemotherapy, hormone, and biological and radiation therapy. The adverse effects observed were nausea, vomiting, weight gain, hair loss and an increased chance of infection.
Multidrug resistance (MDR) is one of the major challenges in treating cancer. Chemotherapy is the most common treatment for BC, statistical data shows that over 90% mortality of cancer patients is attributed to the failed conventional chemotherapeutic drugs due to drug resistance. ATP-binding cassette (ABC) proteins such as P-glycoprotein (P-gp) is responsible for the protection of the cancer cells from high concentration of cytotoxic drugs due to the elevated efflux of the drugs from the cancer cells.
P-gp is highly expressed on the surface of the endothelial cells of the cancer cells which contributes to the lower penetration of cytotoxic drugs. Because of that, the conventional cytotoxic drugs have a very low efficacy and least favorable when the cancer cells became resistance.
Flavonoid is one of the emerging and promising weapons in combating and preventing cancer and anti-inflammatory diseases. In in-vitro and in-vivo studies, flavonoids exhibit anti-proliferation, anti-metastatic and immuno-modulatory properties.
These unique flavonoids qualities suggest new ways to alter tumor signaling, get around chemo-resistance and retrain the tumor micro-environment. Luteolin (Lut) is a flavonoids. Lut is frequently present in fruits, vegetables, and medicinal herbs such as green pepper, celery, broccoli, and parsley. Recent research has shown that Lut has a variety of biological effects, primarily because of its antioxidant and free radical-scavenging properties, including anti-inflammation, anti-allergy, and anticancer.
Furthermore, Lut has been found to reverse MDR and desensitize chemo-resistance cells when co-administer with other chemotherapeutic drugs such as oxaliplatin. Samy et al. reported that Lut causing cytotoxicity in cancer cells, but not on normal cells. However Lut has some limitations, like poor solubility, poor bioavailability and low oral absorption, rendering Lut from its full potential in clinical application.
Therefore, it’s important for researchers in the field to find a solution to its limitation. Poloxamer 407 (Pol) and D-a-tocopheryl polyethylene glycol 1000 succinate (TPGS) enhance the solubility and bioavailability poorly water soluble compounds by encapsulating the compound and loaded it into the hydrophobic cargo. Furthermore, TPGS can reverse MDR by binding to the ATP of P-gp and reduce drug efflux.
The combination of these Lut, Pol and TPGS can affect synergistically towards cancer cells and reversing MDR, in theory. It can be seen to be a perfect alternatives and solution for MDR BC treatment. The aim of the authors is to optimize, develop and characterize the novel drug delivery system to ensure that the micelle have a good EE and size for efficacy and specificity of passive targeting of the drug towards cancer cells.
The optimized micelle was found to have 90% EE with a particle size of less than 40 nm, which was achieved using a 10% concentration of surfactants at a TPGS/Pol ratio of 3:1. The optimized temperature for hydrating the micellar film was 40°C, the optimized mixing time was 1 hour, and the optimized freezing temperature was -80°C.
The solubility of Lut-loaded micelles increased up to 459-fold compared to pure Lut in water. The CMC of TPGS/Pol micelle was 0.001 mg/ml, and the release study showed that Lut-loaded micelles exhibited sustained release behavior. They studied several factors in order to obtain a well-optimized micelle.
After review I suggest accepting for publication at the Journal. The content of the MS is well investigated, well developed, well written, it will good for the audience and Journal.
Dear Editor
After review I found accepting for publication at the Journal. The Investigation is well developed, the results are clearly presented, the content will be good for the Journal and audience. Sincerely yours,
Blas Lotina-Hennsen, PhD
Author Response
Thank you for the comments. all the comments have been properly addressed.

Reviewer 2 Report
Muhammad et al., in this manuscript titled "Optimisation of Luteolin-Loaded TPGS/Poloxamer 407 Nanomicelle: Effect of Copolymers, Hydration Temperature and Duration, and Freezing Temperature on Encapsulation Efficiency, Particle Size, and Solubility" have developed and optimise TPGS/Pol micelles loaded with Luteolin by investigating certain factors that can affect the encapsulation efficiency (EE) and particle size (PS) of the micelle.
Although the manuscript is well-written and scientifically sound. I have few concerns herein to make:
1) Figure 3: The stretching bands in the FT-IR spectra should be labelled and provide more high-quality figure.
2) Figure 4: Authors should provide more clearer XRD spectra and peaks should have hkl values.
3) In vitro word should be italicized.
4) Simple summary section is not necessary.
Minor corrections are required.
Author Response

(The authors gave the same response as above.)

Reviewer 3 Report
1. 1) Abstract is too short and does not clearly express the summary of the work, please correct it. Highlights the significant results. Do not use abbreviation in the abstract like MDR, what does it mean?!!
1. 2) How do you know the combination of Luteolin with vitamin E TPGS (TPGS) and poloxamer (Pol) have synergistic effect to enhance tumor apoptosis and P-glycoprotein inhibition? You did not do any experiment !!! how do you confirm? Please explain. You must to do some invitro and invivo experiment to prove it.
4 How do you confirm your synthesized nano micelle it is not toxic for normal cell.
5. Throughout the text, the language must be adequately check. Apart from formal grammar errors, please check with native English person
6. The quality of the figures must be improving especially the TEM image
7. The results in the table must be statistically analyze? are you results significant or not? What is the unit? For example, encapsulation efficacy and particle size?
8. The introduction and discussion must be explained more specifical. Please cite these papers and improve this section for instance you need to talk more about the role of natural bioactive compounds in cancer and also compare the different carrier system and highlight why you use nano micelle? Discussion needs to be improved. Authors are advised to investigate literature and make a broader comparison between their results and earlier/current reports
• Hepatoprotective effect of nanoniosome loaded Myristica fragrans phenolic compounds in mice-induced hepatotoxicity [DOI: 10.1111/jcmm.17581]
• Antiproliferation effects of nanophytosome‑loaded phenolic compounds from fruit of Juniperus polycarpos against breast cancer in mice model: synthesis, characterization and therapeutic effects
• Synthesized chrysin-loaded nanoliposomes improves cadmium-induced toxicity in mice [https://doi.org/10.1007/s11356-020-10113-7]
• Nanoliposome-Loaded Phenolics from Nasturtium officinale Improves Health Parameters in a Colorectal Cancer Mouse Model
· Satureja khuzistanica Essential Oil-Loaded Solid Lipid Nanoparticles Modified With Chitosan-Folate: Evaluation of Encapsulation Efficiency, Cytotoxic and Pro-apoptotic Properties. https://doi.org/10.3389/fchem.2022.904973
· Borago officinalis L. flower: a comprehensive study on bioactive compounds and its health-promoting properties
· Encapsulated phenolic compounds from Ferula gummosa leaf: A potential phytobiotic against Campylobacter jejuni infection [DOI: 10.1111/jfpp.16802]
Throughout the text, the language must be adequately check. Apart from formal grammar errors, please check with native English person
Author Response

(The authors gave the same response as above.)

Reviewer 4 Report
Muhammad Redza Fahmi Mod Razif and co-authors have presented a study focuses on the design of the Luteolin-loaded TPGS/Poloxamer 407 nano-micelles to enhance the solubility and delivery capacity of the Luteolin. Unfortunately, the manuscript has some serious flaws and does not meet the standards of the journals. The main concerns are:
1. The differences in diameters of empty and loaded micelles have not been established properly.
2. The data collected has failed to show the proper location of the drug within the micelles.
3. In the release kinetics, mathematical models should be employed for the better understandings. Moreover, the standard deviation has not applied which questioned its reproducibility.
4. No cell lines studies included, authors could not define the safety, in-vitro toxicity.
Minor points:
1. Please mention the dilution factor applied in the particles size measurements.
2. The zeta potential measurements could be an interesting approach to define the colloidal stability of the systems.
3. Quality of TEM image and XRD data need to be improved.
4. The peaks of FTIR should be highlighted in the Figure 3.
Minor editing is needed.
Author Response

(The authors gave the same response as above.)

Reviewer 5 Report
In this article, Luteolin-loaded micelle of TPGS and poloxamer was prepared and optimised using film hydration method. This study suggests that this developed mixed micelle can be used to solubilise hydrophobic anti-cancer compound, verifying a nice drug carrier for the potential application. The manuscript is well-organized and it can be published after minor revision.
1. In the introduction, BC is mostly treated with surgery, chemotherapy, hormone, bio-logical and radiation therapy. Recently, phototherapy is also applied in BC, the following literature might be useful, DOI:10.1039/d0cc08162k
2. For Preparation of Luteolin-Loaded Micelle, Lut was mixed together with TPGS and Pol using ethanol as the solvent until the solution becomes homogenous. Please explain why not use water as the solvent.
3. Please be careful for the space between unit and number, for example, 350nm should be 350 nm.
4. The pure Lut sample shows the main characteristic bonds at 3418 cm-1 (strong -OH stretching). Does this peak come from the -OH in water?
5. For In-Vitro Drug Release Study, please explain why Tween 80 were used as release medium.
Author Response

(The authors gave the same response as above.)

Reviewer 6 Report
The manuscript is very interesting, but Authors should improve some parts:
1.why the solubility tests was carried out after 72 hours? Is not too long? Why not to notice the difference?
2. In the section 2.13 Authors should add pH of PBS
3. in my opinion 1 figure of TEM is not enough
4. In the figure 5 all test should be carried out in the same time
5. there are many editorials mistakes.
6. in my opinion the Authors could performed anticancer activity studies
Author Response

(The authors gave the same response as above.)

Round 2
Reviewer 3 Report
I could not find the answers. Please respond all the comments point to point and highlighted in the manuscript.
I could not find the answers. Please respond all the comments point to point and highlighted in the manuscript.
Reviewer 4 Report
All comments are justified.
Author Response
Thank you for the comments.
Round 3
Reviewer 3 Report
1. Abstract is too short and does not clearly express the summary of the work, please correct it. Highlights the significant results. Do not use abbreviation in the abstract like MDR, what does it mean?!!
2. How do you know the combination of Luteolin with vitamin E TPGS (TPGS) and poloxamer (Pol) have synergistic effect to enhance tumor apoptosis and P-glycoprotein inhibition? You did not do any experiment !!! how do you confirm? Please explain. You must to do some invitro and invivo experiment to prove it.
3. How do you confirm your synthesized nano micelle it is not toxic for normal cell.
4. Throughout the text, the language must be adequately check. Apart from formal grammar errors, please check with native English person
5. The quality of the figures must be improving especially the TEM image
6. The results in the table must be statistically analyze? are you results significant or not? What is the unit? For example, encapsulation efficacy and particle size?
7. The introduction and discussion must be explained more specifical. Please cite these papers and improve this section for instance you need to talk more about the role of natural bioactive compounds in cancer and also compare the different carrier system and highlight why you use nano micelle? Discussion needs to be improved. Authors are advised to investigate literature and make a broader comparison between their results and earlier/current reports
• Hepatoprotective effect of nanoniosome loaded Myristica fragrans phenolic compounds in mice-induced hepatotoxicity [DOI: 10.1111/jcmm.17581]
• Antiproliferation effects of nanophytosome‑loaded phenolic compounds from fruit of Juniperus polycarpos against breast cancer in mice model: synthesis, characterization and therapeutic effects
• Synthesized chrysin-loaded nanoliposomes improves cadmium-induced toxicity in mice [https://doi.org/10.1007/s11356-020-10113-7]
• Nanoliposome-Loaded Phenolics from Nasturtium officinale Improves Health Parameters in a Colorectal Cancer Mouse Model
· Satureja khuzistanica Essential Oil-Loaded Solid Lipid Nanoparticles Modified With Chitosan-Folate: Evaluation of Encapsulation Efficiency, Cytotoxic and Pro-apoptotic Properties. https://doi.org/10.3389/fchem.2022.904973
· Borago officinalis L. flower: a comprehensive study on bioactive compounds and its health-promoting properties
· Encapsulated phenolic compounds from Ferula gummosa leaf: A potential phytobiotic against Campylobacter jejuni infection [DOI: 10.1111/jfpp.16802]

Author Response
Reviewer 3
- Abstract is too short and does not clearly express the summary of the work, please correct it. Highlights the significant results.Do not use abbreviation in the abstract like MDR, what does it mean?!!
Reply: Abstract has been improved by inserting significant results into it. Abbreviation has been removed. The new addition part is highlighted in grey.
- How do you know the combination of Luteolin with vitamin E TPGS (TPGS) and poloxamer (Pol) have synergistic effect to enhance tumor apoptosis and P-glycoprotein inhibition? You did not do any experiment !!! how do you confirm? Please explain. You must to do some invitro and invivo experiment to prove it.
Reply: In the manuscript, we have stated the claim (“The combination of these Lut, Pol and TPGS might have synergistic effects towards cancer cells and reversing MDR, in theory”) was only an assumption or theory as a base reason on why we used this combination. The theory presented was made based on previous studies. Vitamin E TPGS has been proven previously in reversing MDR by inhibiting P-glycoprotein besides having cell apoptosis effect towards cancer cells (Guan et al., 2020). Poloxamer 407 has been proven to give cell apoptosis effect on cancer cells despite having no or minimal effect on P-glycoprotein inhibition (Pitto-Barry & Barry, 2014; Yu et al., 2021). Luteolin has a wide range of anti-cancer properties and also can be used to desensitize MDR cancer cells (Huang et al., 2019; Ye et al., 2019). From these evidence, we assumed that this combination can increase the anti-cancer effect on MDR cancer cells. We are well aware that we have to prove it by doing in-vitro and in-vivo experiments, we have planned on doing these experiments too. Since this manuscript more focused on the formulation part, the in-vitro and in-vivo study will be reported in the next manuscript.
Bachu, R. D., Chowdhury, P., Al-Saedi, Z. H. F., Karla, P. K., & Boddu, S. H. S. (2018). Ocular Drug Delivery Barriers—Role of Nanocarriers in the Treatment of Anterior Segment Ocular Diseases. In Pharmaceutics. https://doi.org/10.3390/pharmaceutics10010028
Gendrisch, F., Esser, P. R., Schempp, C. M., & Wölfle, U. (2020). Luteolin as a Modulator of Skin Aging and Inflammation. In Biofactors. https://doi.org/10.1002/biof.1699
Guan, Y., Wang, L. yan, Wang, B., Ding, M. hong, Bao, Y. ling, & Tan, S. wei. (2020). Recent Advances of D-α-tocopherol Polyethylene Glycol 1000 Succinate Based Stimuli-responsive Nanomedicine for Cancer Treatment. Current Medical Science, 40(2), 218–231. https://doi.org/10.1007/s11596-020-2185-1
Huang, L., Jin, K., & Lan, H. (2019). Luteolin inhibits cell cycle progression and induces apoptosis of breast cancer cells through downregulation of human telomerase reverse transcriptase. Oncology Letters, 17(4), 3842–3850. https://doi.org/10.3892/ol.2019.10052
Pitto-Barry, A., & Barry, N. P. E. (2014). Pluronic® block-copolymers in medicine: From chemical and biological versatility to rationalisation and clinical advances. Polymer Chemistry, 5(10), 3291–3297. https://doi.org/10.1039/c4py00039k
Ruiz-Moreno, C., Jimenez-Del-Rio, M., Sierra-Garcia, L., Lopez-Osorio, B., & Velez-Pardo, C. (2016). Vitamin E Synthetic Derivate—TPGS—selectively Induces Apoptosis in Jurkat T Cells via Oxidative Stress Signaling Pathways: Implications for Acute Lymphoblastic Leukemia. In Apoptosis. https://doi.org/10.1007/s10495-016-1266-x
Yang, C., Wu, T., Qi, Y., & Zhang, Z. (2018). Recent advances in the application of vitamin E TPGS for drug delivery. Theranostics, 8(2), 464–485. https://doi.org/10.7150/thno.22711
Ye, Q., Liu, K., Shen, Q., Li, Q., Hao, J., Han, F., & Jiang, R. W. (2019). Reversal of multidrug resistance in cancer by multi-functional flavonoids. Frontiers in Oncology, 9(JUN), 1–16. https://doi.org/10.3389/fonc.2019.00487
Yu, J., Qiu, H., Yin, S., Wang, H., & Li, Y. (2021). Polymeric drug delivery system based on pluronics for cancer treatment. Molecules, 26(12), 1–23. https://doi.org/10.3390/molecules26123610
- How do you confirm your synthesized nano micelle it is not toxic for normal cell.
Reply: Luteolin is safe for human consumption as it is a flavonoid in natural plants. Study also showed that luteolin is safe to human cell and selectively exhibit cytotoxicity towards cancer cell (Gendrisch et al., 2020). Vitamin E TPGS has selectively cytotoxicity towards cancer cell, but not towards normal cell (Bachu et al., 2018; Ruiz-Moreno et al., 2016) and Poloxamer are also cytotoxic towards cancer cells and minimal effect on normal cells (Yang et al., 2018)
Bachu, R. D., Chowdhury, P., Al-Saedi, Z. H. F., Karla, P. K., & Boddu, S. H. S. (2018). Ocular Drug Delivery Barriers—Role of Nanocarriers in the Treatment of Anterior Segment Ocular Diseases. In Pharmaceutics. https://doi.org/10.3390/pharmaceutics10010028
Gendrisch, F., Esser, P. R., Schempp, C. M., & Wölfle, U. (2020). Luteolin as a Modulator of Skin Aging and Inflammation. In Biofactors. https://doi.org/10.1002/biof.1699
Ruiz-Moreno, C., Jimenez-Del-Rio, M., Sierra-Garcia, L., Lopez-Osorio, B., & Velez-Pardo, C. (2016). Vitamin E Synthetic Derivate—TPGS—selectively Induces Apoptosis in Jurkat T Cells via Oxidative Stress Signaling Pathways: Implications for Acute Lymphoblastic Leukemia. In Apoptosis. https://doi.org/10.1007/s10495-016-1266-x
Yang, C., Wu, T., Qi, Y., & Zhang, Z. (2018). Recent advances in the application of vitamin E TPGS for drug delivery. Theranostics, 8(2), 464–485. https://doi.org/10.7150/thno.22711
- Throughout the text, the language must be adequately check. Apart from formal grammar errors, please check with native English person.
Reply: The revised manuscript has been proofread by a native English colleague from Centre of Foundation and Languages Studies.
- The quality of the figures must be improving especially the TEM image.
Reply: We have submitted the original TEM image from the software. This is the best quality image that we have.
- The results in the table must be statistically analyze? are you results significant or not? What is the unit? For example, encapsulation efficacy and particle size?
Reply: We have included the statistical analysis into the tables. The unit also has been included into the tables. Unit for EE is (% w/w), unit for particle size (nm) and unit for concentration (% w/v). The statistical analysis is described under 2.14 as highlighted in grey. The statistical results can be found in table 1-3 as highlighted in grey.
- The introduction and discussion must be explained more specifical. Please cite these papers and improve this section for instance you need to talk more about the role of natural bioactive compounds in cancer and also compare the different carrier system and highlight why you use nano micelle? Discussion needs to be improved. Authors are advised to investigate literature and make a broader comparison between their results and earlier/current reports
- Hepatoprotective effect of nanoniosome loaded Myristica fragrans phenolic compounds in mice-induced hepatotoxicity [DOI: 10.1111/jcmm.17581]
- Antiproliferation effects of nanophytosome‑loaded phenolic compounds from fruit of Juniperus polycarpos against breast cancer in mice model: synthesis, characterization and therapeutic effects
- Synthesized chrysin-loaded nanoliposomes improves cadmium-induced toxicity in mice [https://doi.org/10.1007/s11356-020-10113-7]
- Nanoliposome-Loaded Phenolics from Nasturtium officinale Improves Health Parameters in a Colorectal Cancer Mouse Model
- Satureja khuzistanica Essential Oil-Loaded Solid Lipid Nanoparticles Modified With Chitosan-Folate: Evaluation of Encapsulation Efficiency, Cytotoxic and Pro-apoptotic Properties. https://doi.org/10.3389/fchem.2022.904973
- Borago officinalis L. flower: a comprehensive study on bioactive compounds and its health-promoting properties
- Encapsulated phenolic compounds from Ferula gummosa leaf: A potential phytobiotic against Campylobacter jejuni infection [DOI: 10.1111/jfpp.16802]
Reply: We have included all these references in our revised manuscript. We have improved the discussion based on the suggestion.
NOTES: All of the changes we have made in the manuscript have been highlighted using the grey highlight

Round 4
Reviewer 3 Report
The authors completely done the comments and manuscript has been accepted in the present form.